



# Warming and ocean acidification may decrease estuarine dissolved organic carbon export to the ocean

Michelle N. Simone[1], Kai G. Schulz[1], Joanne M. Oakes[1], Bradley D. Eyre[1]

[1]Centre for Coastal Biogeochemistry, School of Environment, Science and Engineering, Southern Cross University, Lismore, NSW, 2480, Australia

*Correspondence to*: Michelle N. Simone (mnhsimone@gmail.com)

**Abstract.** Estuaries make a disproportionately large contribution of dissolved organic carbon (DOC) to the global carbon cycle, but it is unknown how this will change under a future climate. As such, the response of DOC fluxes from microbially dominated unvegetated sediments to individual and combined future climate stressors of warming (from $\Delta$-3 °C to $\Delta$+5 °C on ambient mean temperatures) and ocean acidification (OA, ~2 times the current partial pressure of $CO_2$, $pCO_2$) was investigated ex situ. Warming alone increased sediment heterotrophy, resulting in a proportional increase in sediment DOC uptake, with sediments becoming net sinks of DOC (3.5 to 8.8 mmol-C m$^{-2}$ d$^{-1}$) at warmer temperatures ($\Delta$+3 °C and $\Delta$+5 °C, respectively). This temperature response changed under OA conditions, with sediments becoming more autotrophic and a greater sink of DOC (1 to 4 times greater than under current-$pCO_2$). This response was attributed to the stimulation of heterotrophic bacteria with the autochthonous production of labile organic matter by microphytobenthos. Extrapolating these results to the global area of unvegetated subtidal estuarine sediments, the future climate of warming ($\Delta$+3 °C) and OA may decrease the estuarine export of DOC by ~80 % (~150 Tg-C yr$^{-1}$) and have a disproportionately large impact on the global DOC budget.

## 1    Introduction

The aquatic dissolved organic carbon (DOC) pool is one of the largest pools of organic carbon on earth  (Hedges, 1987) and roughly equivalent in size to the atmospheric $CO_2$ reservoir (Siegenthaler and Sarmiento, 1993). The role of DOC in the long-term storage of carbon in the ocean has been a focus of research for decades (Siegenthaler and Sarmiento, 1993; Hansell et al., 2009; Bauer and Bianchi, 2011; Wagner et al., 2020), with DOC reaching the ocean interior being effectively stored for millennia (Hansell et al., 2009). Although phytoplankton in the surface ocean are the main source of DOC globally, with an estimated production of around 50 Pg-C yr$^{-1}$, only 0.3 % of the DOC produced by phytoplankton reaches the ocean interior (Hansell et al., 2009), with most of the DOC rapidly remineralised by heterotrophic bacteria in the water column (Azam, 1998). Only more recently has the coastal zone been considered a major source of DOC export to the open ocean and deep-sea (Duarte et al., 2005; Maher and Eyre, 2010; Krause-Jensen and Duarte, 2016). The shallow coastal zone accounts for 1 to 10 % of global net primary production (NPP) (Duarte and Cebrián, 1996), and up to 33 % of the associated DOC is exported offshore and stored in the ocean interior (Krause-Jensen and Duarte, 2016).



Although shallow estuaries and fringing wetlands make up only ~22 % of the world's coastal area (Costanza et al., 1997) and 8.5 % of the total marine area (Costanza et al., 1997) they are quantitatively significant in terms of DOC processing and offshore transport (Smith and Hollibaugh, 1993). The quantity of DOC reaching the ocean interior from coastal areas is up to 3.5 times more than that derived from production in the surface ocean. This is despite coastal areas having a DOC production rate only 0.2 to 3.9 % that of the open ocean (Duarte, 2017). As such, small changes to the coastal production and export of

DOC may have a disproportionate influence on the global DOC budget.

Euphotic estuarine sediments occupy the coastal boundary between terrestrial and marine ecosystems. Microalgal communities (microphytobenthos, or MPB) are ubiquitous in these sediments, occupying ~40 to 48 % of the coastal surface area (Gattuso et al., 2020), and generating up to 50 % of total estuarine primary productivity (Heip et al., 1995; MacIntyre et al., 1996; Underwood and Kromkamp, 1999). MPB exude some of the carbon they fix as extracellular substances, including

carbohydrates (Oakes et al. 2010), and can therefore be a source of relatively labile DOC in net autotrophic sediments (Cook et al., 2004; Oakes and Eyre, 2014; Maher and Eyre, 2010). The dominant sink of DOC in estuarine sediments, however, is microbial mineralisation by heterotrophic bacteria (Azam, 1998). These heterotrophic bacteria not only consume autochthonous DOC (Boto et al., 1989), but their biomass is influenced by the lability of sediment organic matter (OM) lability (Hardison et al., 2013), which can be directly linked to and stimulated by the MPB productivity of the sediments (Hardison et

al., 2013; Cook et al., 2007). Estuarine sediments are therefore a potentially important sink for DOC.

Although estuarine sediments can be a source of autochthonous DOC to the ocean (Duarte, 2017; Krause-Jensen and Duarte, 2016; Maher and Eyre, 2010), they also control the quantity and quality of allochthonous and terrigenous DOC (tDOC) that passes through them (Fichot and Benner, 2014). tDOC supports heterotrophy in estuaries and unvegetated coastal zones, which are often sources of $CO_2$ to the ocean (Frankignoulle et al., 1998; Fichot and Benner, 2014). Much of the tDOC is efficiently

mineralised in estuaries before it reaches the open ocean (Opsahl and Benner, 1997). The disproportionate contribution of estuaries to the export of DOC to offshore marine ecosystems, relative to their surface area requires a better understanding of how the balance of DOC sources and sinks within estuaries may change in the future when exposed to a high-$CO_2$ climate of increased temperatures and ocean acidification (OA).

Climate projection models suggest that atmospheric $CO_2$ concentrations could more than double by the end of the century,

increasing the partial pressure of $CO_2$ ($pCO_2$ = +580 $\mu$atm), decreasing pH (~0.3 units, OA), and increasing temperature ($\Delta$+2 to 4 °C) in the surface ocean (RCP8.5; IPCC, 2013). Individually, increased temperature and $CO_2$ have been reported to enhance primary productivity and DOC production in arctic (Engel et al., 2013; Czerny et al., 2013) and temperate phytoplankton communities (Wohlers et al., 2009; Engel et al., 2011; Liu et al., 2017; Novak et al., 2018; Taucher et al., 2012), and temperate stream sediments (Duan and Kaushal, 2013). However, one study in a temperate fjord reported no enhancement

of DOC production despite $CO_2$ enhanced phytoplankton productivity (Schulz et al., 2017). This uncertainty of response to individual climate stressors is exacerbated when considering how the combination of OA and warming will affect DOC processing. To date, only one study has considered this combined stressor effect on DOC (Sett et al., 2018), observing no



difference in DOC production by temperate phytoplankton relative to current conditions under the combined stressors (Sett et al., 2018).

To understand the effect of future climate on DOC fluxes under future climate conditions, it is essential that we consider both individual and combined effects of OA and warming. Here we focus on the changes in DOC fluxes in unvegetated estuarine sediments, as these systems have the potential to take up significant portions of DOC that is currently exported to the coastal ocean. In this study, benthic DOC responses in unvegetated estuarine sediments were investigated over an 8 °C temperature range under both current and future-$p$CO$_2$ conditions in an ex situ laboratory incubation.

We expected that warming would promote a stronger heterotrophic, than autotrophic, microbial response (Patching and Rose, 1970; Vázquez-Domínguez et al., 2012; Yang et al., 2016), and as such, there would likely be more DOC remineralisation (Lønborg et al., 2018) than 'new' DOC production (Wohlers et al., 2009; Engel et al., 2011; Novak et al., 2018). Moreover, despite the potential stimulation by OA of primary productivity in unvegetated muddy sediments (Vopel et al., 2018), and potential enhancement of DOC production (Engel et al., 2013; Liu et al., 2017), this labile DOC may promote bacterial

productivity and DOC mineralisation (Hardison et al., 2013). In addition, increased DOC availability alone may increase heterotrophic bacterial biomass production and activity (Engel et al., 2013). We therefore predicted that increases in DOC production from OA alone or in combination with warming may be counteracted by increased consumer activity, depleting the available DOC pool under future climate conditions.

## 2 Methods

### 2.1 Study site

A subtidal site (~1.5 m below mean sea level) in the subtropical Clarence River Estuary, Australia, was used for this study (29°24.21'S, 153°19.44'E; Figure 1). Sediment at the site was unvegetated and characterised as a cohesive sandy mud (31-36 % grains 250-500 $\mu$m, 61-65 % 63-250 $\mu$m, and ~2% <63 $\mu$m, Lewis and McConchie, 1994). Temperature ± 0.3 °C, pH ± 0.5 units, and salinity (± <1 %) were measured over 24 hours using a Hydrolab (HL7) submerged at the site. The tidal cycle

introduced a salinity range of 10-35, pH range of 7.42-8.15 units (min-max), and mean daily temperature of 23.9 ± 1.6 °C (20-25 °C). The surface sediments (0-2 cm) had a porosity of 0.43 and an organic matter content of ~3.5 % (of dry weight), determined from mass loss after combustion (490 °C) of dried sediment (60 °C) (Luczak et al., 1997). The Clarence River Estuary has low nutrient loading (Eyre and Pont, 2003) with dissolved inorganic nitrogen (DIN) concentrations <2 $\mu$M (Eyre, 2000). This is consistent with concentrations determined at the time of this study (~0.9-1.9 $\mu$M DIN).

### 2.2 Core collection

Sediment (~20 cm depth) was collected and capped in acrylic cores (9 cm diameter x 47 cm length) allowing for ~1.8 L of overlying water on the 9th (33 cores) and 16th (27 cores) of January 2018. Thalassinidean shrimp, *Trypaea australiensis*, burrows were avoided and therefore excluded from the collected cores as their occasional inclusion would result in considerable variability in sediment processes (Webb and Eyre, 2004) that would mask potential treatment effects. To ensure





sediments were subtidal, cores were collected during low tide. Immediately after core collection, ~700 L of site water was also
collected to fill a laboratory incubation setup.

## 2.3    Incubation setup

Within 6 hours of core collection, all cores were in the laboratory, submerged, uncapped in site water. The cores were placed
in 1 of 4 temperature tanks, Control (23 °C), Δ-3 °C (20 °C), Δ+3 °C (25 °C), and Δ+5 °C (28 °C) filled with ~80 L of site
water, with temperatures maintained and monitored via thermo-regulating aquarium pumps. Each tank had two sets of 3 cores
(n = 6) except for the Control tank, which had an additional 3 cores (n = 9) for background isotope determination.

The ex situ study design allowed control of temperature, $p$CO$_2$ and light that would be difficult to achieve in situ. Due to
limited space, this investigation was conducted over two weeks with two complementary incubations repeated back-to-back.
The incubation in the first week (January 9-12, 2018) had cores in the 4 temperature tanks subjected to future-$p$CO$_2$ (~1000
$\mu$atm), achieved with a CO$_2$ enriched airstream (initially adjusted and set when attached to a LICOR (LI-7000)) bubbled into
tank water via airstones and air pumps to simulate the future atmospheric CO$_2$ condition (~1000 $\mu$atm; RCP8.5, Collins et al.,
2013), whereas the incubation in the second week (January 16-20, 2018) maintained current-$p$CO$_2$ (~450 $\mu$atm) by circulating
ambient laboratory air through the tank water via airstones and air pumps. An additional tank was included in the future-$p$CO$_2$
incubation. This tank was a control tank equivalent to the control tank present in the current-$p$CO$_2$ incubation, allowing for
comparison of the two separate incubations (see Table 1 for details).

Water columns within cores were stirred at ~60 rpm throughout the incubations via magnetic stir bars (~5 cm above sediment
surface) interacting with an external rotating magnet, ensuring water columns were well mixed whilst avoiding sediment
disturbance (Ferguson et al., 2003, 2004). High pressure sodium lamps (400 W; PHILIPS Son-T Argo 400) were used to
simulate mean daytime field conditions, providing ~270-280 $\mu$mol quanta m$^{-2}$ s$^{-1}$ of photosynthetically active radiation (PAR)
at the water surface of the tanks. Lamps were turned on in the mornings in line with natural diel light cycling, following a
similar in situ ~12:12 hour dark:light cycle. After 24-48 hour of cores preincubating at treatment conditions, cores were capped
for a short-term 20 hour (10:10 hour, dark:light) solute-flux incubation to measure rates of O$_2$, DIC, and DOC production and
consumption over a diel cycle. The temperature manipulations remained within 12 % of their in situ ranges (see Sect. 2.1) to
reduce any potential shock effect that may be experienced by the sediment community in a short-term incubation. The two day
pre-incubation would be sufficient for up to three or six generations of the dominant microbial members of unvegetated
estuarine sediments, diatoms and cyanobacteria, respectively (Mori et al., 1996; Greene et al., 1992), allowing time for the
microbial community to acclimatise to the new conditions.

### 2.3.1    Solute flux incubation

Carbon fluxes were measured over a 20 hour period from three cores from each tank. To adhere to natural diel cycling, cores
were capped at dusk to start the incubation on a dark cycle. Samples were collected at three time points in the diel cycle (dark
start (dusk), dark end/light start (dawn), and light end (dusk)). Water was collected and syringe-filtered to determine
concentrations of DIC (0.45 $\mu$m Minisart filter, 100 ml serum bottle; without headspace, poisoned with 50 $\mu$l of saturated



$HgCl_2$, stored at room temperature) and DOC (GF/F filter, 40 ml glass vial with silicon septum; without headspace, poisoned with 20 $\mu$l of $HgCl_2$, injected with 200 $\mu$l of 85 % $H_3PO_4$, stored at room temperature). As water was removed for sampling it

was replaced with gravity-fed water maintained in a collapsible bag under the same atmospheric conditions and temperature. After all cores were sampled, dissolved oxygen (DO) concentrations, temperature, and pH were measured using a high precision Hach HQ40d Multiprobe meter with an LDO-probe and pH-probe, calibrated to 3-point NIST buffer scale ($R^2$ = 0.99). Probes were inserted into a resealable port fitted in each lid, ensuring no incubation water exchanged with tank water. After the dawn sampling (time point 3), lamps were switched on.

DIC concentrations were determined with an AIRICA system (MARIANDA, Kiel) via infrared absorption using a LI-COR LI-7000, and corrected for accuracy against certified reference material, batch #171 (Dickson, 2010). Measurements on four analytical replicates of 1.5 ml sample volume were used to calculate DIC concentration as the mean of the last three out of four measurements (typical overall uncertainty, <1.5 $\mu$mol kg$^{-1}$). DIC and pH measurements were then used to calculate the remaining carbonate chemistry parameters (Table 1) using $CO_2$Sys (Pierrot et al., 2006). Total borate concentrations

(Uppström, 1974) and boric acid (Dickson, 1990) and stoichiometric equilibrium constants for carbonic acid (Dickson and Millero, 1987), refit from Mehrbach et al. (1973), were used. A comparison of measured pH (free scale) with a Hach HQ40d Multiprobe meter and calculated pH using measured total alkalinity and DIC (Table S1), indicated an uncertainty of $\pm$ 0.05 pH units for potentiometric pH measurements without synthetic seawater buffers. Assuming the same uncertainty in pH measurements in this study and propagating it with the uncertainty of DIC, this translates to a $p$CO$_2$ uncertainty of $\pm$ ~110 and

~56 $\mu$atm under future and current-$p$CO$_2$, respectively. This uncertainty is well within the treatment variability measured among cores (Table 1) and is therefore considered unlikely to have contributed substantially to differences in treatment response. DOC concentrations were measured via continuous-flow wet-oxidation using an Aurora 1030W total organic carbon analyser (Oakes et al., 2011) (uncertainty of ~3 %).

## 2.4    Data analysis

The dissolved oxygen and DIC measurements were used to estimate benthic microalgal production inside the cores. Net primary production and respiration (NPP and R, $\mu$mol-O$_2$ m$^{-2}$ h$^{-1}$) were defined as the light or dark cycle oxygen evolution, respectively, where DIC and DOC light and dark fluxes ($\mu$mol-C m$^{-2}$ h$^{-1}$) were defined using the evolution of DIC and DOC concentrations, respectively. Flux (NPP, R, DIC, or DOC) was calculated as:

$$\text{Flux} = \frac{(\text{End} - \text{Start}) \times \text{V}}{(\text{T} \times \text{A})} \qquad\qquad \text{Eq. (1)}$$

where End and Start are the dissolved oxygen, inorganic carbon, or organic carbon concentrations ($\mu$mol-O$_2$ or -C L$^{-1}$) at the end and start of the light or dark cycle, V is the water column volume (L), T is hours of incubation, and A is surface area of the core.

Gross primary productivity (GPP, $\mu$mol-O$_2$ m$^{-2}$ h$^{-1}$) was calculated using NPP/R.

$$\text{GPP} = -\text{R} + \text{NPP} \qquad\qquad \text{Eq. (2)}$$





The production to respiration ratio (P/R) was calculated using GPP and R rates scaled for a 12:12 hour light:dark diel cycle

(Eyre et al., 2011).

$$P/R = \frac{(GPP) \times 12hr}{(-R \times 24hr)} \qquad \text{Eq. (3)}$$

Finally, net fluxes for DIC and DOC were calculated from the dark and light fluxes from Eq. (1) and presented as mmol-C per

$m^2$ per day for a 12:12 hour light:dark diel cycle.

Net flux = ((Dark flux x 12h) + (Light flux x 12h))/1000        Eq. (4)

$Q_{10}$ values were used to evaluate the temperature dependence of metabolic rates to temperature increases of 10 °C. This was

expressed simply as an exponential function:

$$Q_{10} = \left(\frac{R_2}{R_1}\right)^{10°C\big/(T_{opt}-T_1)} \qquad \text{Eq. (5)}$$

where $R_1$ and $R_2$ are the R, NPP, or GPP rates measured at temperatures 20 °C ($T_1$) and optimal temperatures ($T_{opt}$), where rates

are highest, respectively.

### 2.4.1   Scaling rates

Rates in the overlapping control cores each week were checked to ensure comparability between incubations. If means ($\pm$ SD)

were significantly different (did not overlap), rates from individual treatment cores were scaled to the overall mean control

rate of both weeks (n = 6). This was done by calculating the proportion of treatment rates to the control rates present in its

week (n = 3),

$$\text{tProp.} = \frac{tRate}{Control} \qquad \text{Eq. (6)}$$

where tRate is the individual core rate, and tProp. is the proportional core rate from the mean control (Control, n = 3) present

in its week. This proportional rate was then multiplied by the overall control mean rates (n = 6) to scale individual core rates

and calculate comparable treatment means (n = 3) across incubations (see Sect. 3.1 for details on scaled rates).

## 2.5   Statistical analysis

Homogeneity of variances (Levene's test) were tested before analysis to minimize the type I error potential. All tests were run

in MATLAB (Mathworks, 2011) with significance defined at a maximum alpha of < 0.05. Where Levene's test returned a

significant result, datasets were either log transformed or, if negative values were present, an alpha of 0.01 was used for the

following ANOVAs.

### 2.5.1   Net variability with temperature and CO₂

Net fluxes among treatments were compared to identify the individual and combined effects of temperature and $p$CO$_2$ on O$_2$,

DIC, and DOC fluxes. To investigate the effect of increased $p$CO$_2$ alone, data from control temperature cores at both current

and future-$p$CO$_2$ were compared using a paired-sample t-test. One-way analyses of variance (ANOVAs) were run for each

$p$CO$_2$ level to investigate differences in sediment responses across temperatures (n = 4). Post-hoc Tukey's tests were then used





to determine which temperatures had similar or different responses. Finally, a two-way ANOVA was conducted on each dataset
to identify whether there were interacting effects on $O_2$, DIC and DOC fluxes of the combined stressor condition, temperatures
(n = 4) and/or $CO_2$ concentrations (n = 2).

### 2.5.2   Diel variability with temperature for DIC and DOC fluxes

Differences between dark and light cycles were compared to further investigate changes observed in DIC and DOC net
variability. Similar analyses to those described above were applied here. To examine differences among temperatures (n = 4),
light-condition (n = 2) and whether light-condition significantly interacted with temperature response, two-way ANOVAs
were applied to current and future-$p$CO$_2$ cores, separately. Following this, each light-condition was further investigated to
consider the individual temperature responses in the light and dark separately using one-way ANOVAs and Post-hoc Tukey's
tests.

## 3     Results

### 3.1     Overlapping control scaling

Mean rates calculated from overlapping control cores present in each week were compared to establish whether the two sets
of incubations were directly comparable, and whether changes attributed to future-$p$CO$_2$ were truly due to that treatment, and
not just a temporal shift in how the sediments were behaving. The P/R ratios were similar for incubations (0.84 ± 0.01 and
0.83 ± 0.04, respectively), however, the magnitude of the R and NPP fluxes were ~23 % greater for control cores in the future-
$p$CO$_2$ week (Table S2; discussed in Sect. 4.0). As such, R and NPP rates of cores were scaled to mean control rates (n = 6)
using the proportional rate difference calculated between the treatments and the individual controls present in the respective
weeks (n = 3) (Eq. (7)). Scaled rates were within ± 13 % of actual rates. There were no significant differences between controls
for light or dark production of DIC or DOC.

### 3.2     Productivity and respiration responses to OA

Future-$p$CO$_2$ alone (under ambient temperature) significantly increased P/R by ~20 % over control ratios (paired-sample: t =
-14.14, p = 0.005, df = 2; Figure 2d).  This was a result of significant increases in NPP (~42 %) from control conditions (paired-
sample: t = -7.57, p = 0.017, df = 2; Figure 2b), in concert with no significant change in R (paired-sample: t = 2.68, p = 0.12
df = 2; Figure 2a). Similarly, significant increases of DIC uptake in the light reflected the significant increases in NPP with
future-$p$CO$_2$ at ambient temperatures (paired-sample: t = -18.88, p = 0.003, df = 2; Figure 3c). Like R, DIC in the dark did not
change with $p$CO$_2$ (paired-sample: t = 0.32, p = 0.78, df = 2; Figure 3b). GPP also significantly increased with OA at ambient
temperatures (paired-sample: t = -5.70, p = 0.03, df = 2; Figure 2c), with net DIC significantly shifting from a slight efflux to
a slight influx (paired-sample: t = -6.91, p = 0.02, df = 2; Figure 3a).

### 3.3     Productivity and respiration responses to temperature and OA



R, NPP, GPP and P/R had strong responses to temperature with OA only affecting light cycle NPP and in turn, GPP and P/R. R response to temperature was similar at both current and future-$p$CO$_2$ (two-way interaction: $F_{3,16}$ = 0.77, p = 0.53; Figure 2a), with no effect of $p$CO$_2$ on R response (CO$_2$ effect two-way: $F_{1,16}$ = 0.99, p = 0.34; Figure 2a). Accordingly, Q$_{10}$ values between $p$CO$_2$ conditions were similar, 1.66 and 1.69 for current and future-$p$CO$_2$, respectively (Table 2). R increased by ~11 % and ~29 % in higher temperature cores (Δ+3 °C and Δ+5 °C, respectively) and decreased by ~16 % in Δ-3 °C cores (temperature
effect two-way: $F_{3,16}$ = 36.93, p <0.0001; Figure 2a).

NPP response of sediments was significantly affected by the interaction of $p$CO$_2$ and temperature (two-way interaction: $F_{3,16}$ = 8.92, p = 0.001; Figure 2b). Under current-$p$CO$_2$, NPP response was significantly decreased with increased temperature (one-way: $F_{3,8}$ = 41.94, p < 0.0001; Figure 2b), with NPP rates shifting from net autotrophic in low and control temperature cores (efflux of 590 ± 121 and 613 ± 10 $\mu$mol-O$_2$ m$^{-2}$ h$^{-1}$, respectively) to net heterotrophic in higher temperature cores (influx
of 163 ± 228 and 390 ± 97 $\mu$mol-O$_2$ m$^{-2}$ h$^{-1}$, for Δ+3°C and Δ+5°C respectively). Warming therefore resulted in a reduction in rates by 126 % at Δ+3 °C and 164 % at Δ+5 °C, compared to the control (Figure 2a). In contrast, NPP response to temperature under future-$p$CO$_2$ maintained net autotrophy in the light at Δ+3 °C and only resulted in net heterotrophy in the highest temperature treatments (one-way: $F_{3,8}$ = 53.01, p <0.0001; Figure 2b). Although OA in general significantly increased NPP rates over those measured under current-$p$CO$_2$ conditions (CO$_2$ effect, two-way: $F_{1,16}$ = 21.92, p = 0.0003; Figure 2b), and Q$_{10}$
increased from 1.13 to 1.92 (Table 2), the NPP response to OA at Δ+3 °C, reflecting stimulation of primary production, allowed sediments to remain net autotrophic in the light instead of shifting to net heterotrophy as they did under current-$p$CO$_2$ (Figure 2b).

GPP reflected a similar interactive stressor response to that described for NPP (two-way interaction: $F_{3,16}$ = 9.39, p = 0.001; Figure 2c). OA significantly increased GPP rates at ambient and Δ+3 °C temperatures (CO$_2$ effect, two-way: $F_{1,16}$ = 24.77, p
= 0.0001; Figure 2c), resulting in a stronger temperature response of future-$p$CO$_2$ sediments (one-way: $F_{3,8}$ = 40.90, p <0.0001; Figure 2c) than current-$p$CO$_2$ sediments (one-way: $F_{3,8}$ = 16.89, p = 0.001; Figure 2c). This GPP temperature dependence increase was supported by Q$_{10}$ value differences between current and future-$p$CO$_2$ conditions, increasing from 1.46 to 2.27 (Table 2). The differences in P/R among treatments further highlighted significant interaction of temperature and $p$CO$_2$ (two-way interaction: $F_{3,16}$ = 5.86, p = 0.007; Figure 2d), suggesting GPP responses to $p$CO$_2$ were strong enough to alter the overall
productivity of the sediments. Under current-$p$CO$_2$, GPP rates had a slight, but insignificant rate increase from lowered to control temperatures (~12 %), where rates significantly decreased at temperatures higher than control (~45 % and ~50 % for Δ+3 °C and Δ+5 °C, respectively; Figure 2c). As such, P/R reflected GPP with a clear separation between control and Δ-3 °C having a higher P/R (0.84 ± 0.01 and 0.89 ± 0.07, respectively) than the significantly lower ratios (one-way: $F_{3,8}$ = 49.41, p <0.0001; Figure 2d) calculated in increased temperature cores (0.42 ± 0.11 and 0.33 ± 0.05 for Δ+3 °C and Δ+5 °C,
respectively). Similarly, under future-$p$CO$_2$, the effect of GPP on P/R was clear. The positive effect of OA on GPP response pushed the P/R ratio of Δ-3 °C and control temperature cores to ~1 (1.09 ± 0.16 and 1.03 ± 0.03, respectively), suggesting the ecosystem shifted to net autotrophy under those conditions. The positive effect of OA was also highlighted at Δ+3 °C, with





P/R ($0.77 \pm 0.13$) remaining close to current ecosystem ratio ($0.84 \pm 0.01$) instead of significantly dropping like those calculated in $\Delta+5$ °C cores (one-way: $F_{3,8} = 38.58$, p <0.0001; Figure 2d).

## 3.4 DIC fluxes

DIC fluxes mirrored those of dissolved oxygen (Figure 3 and Figure 2) with both light and dark DIC:DO ratios near 1:1 (Figure 4). In the dark, DIC reflected R responses to temperature; like R, DIC responses to temperature did not differ with $p$CO$_2$ (two-way interaction: $F_{3,16} = 0.92$, p = 0.45; Figure 3b) and rates increased with increasing temperature (temperature effect two-way: $F_{3,16} = 12.66$, p = 0.0002; Figure 3b). In the light, there was a significant interactive effect of temperature and $p$CO$_2$ on DIC fluxes (two-way interaction: $F_{3,16} = 12.01$, p = 0.0002; Figure 3c). Under current-$p$CO$_2$, DIC reflected NPP responses to temperature, with DIC taken up at $\Delta$-3 °C and control temperatures and effluxed at $\Delta+3$ °C and $\Delta+5$ °C (one-way: $F_{3,8} = 21.33$ p = 0.0004; Figure 3c).

Net DIC responses were significantly affected by the interaction of $p$CO$_2$ and temperature (two-way interaction: $F_{3,16} = 9.69$, p = 0.001; Figure 3a). Like differences in O$_2$, significant differences between $p$CO$_2$ conditions were also measured in the $\Delta+3$ °C temperature cores. At $\Delta+3$ °C, net DIC production in future-$p$CO$_2$ cores was ~62 % lower than that measured at the same temperature under current-$p$CO$_2$ (paired-sample: t = 5.82, df = 2, p = 0.03; Figure 3a). This again reflected changes in light cycle production, with light DIC effluxes at $\Delta+3$ °C under current-$p$CO$_2$ becoming influxes under future-$p$CO$_2$ ($132 \pm 74$ $\mu$mol-C m$^{-2}$ h$^{-1}$ to $-617 \pm 88$ $\mu$mol-C m$^{-2}$ h$^{-1}$, respectively; Figure 3a).

## 3.5 DOC fluxes

At current-$p$CO$_2$, increasing temperature resulted in a significant shift in net DOC fluxes from effluxes at the two lower temperatures ($\Delta$-3 °C and control) to uptakes at the two higher temperatures at current-$p$CO$_2$ (one-way: $F_{3,8} = 6.96$, p = 0.013; Figure 5a). The light and dark cycle contributions of these net trends at current-$p$CO$_2$ were also affected by temperature (two-way interaction: $F_{3,16} = 13.18$, p = 0.0001; Figure 5b). DOC fluxes in the dark shifted from an efflux at $\Delta$-3 °C to an uptake at control temperature, with higher uptake rates at $\Delta+5$ °C (26 % higher than control rates; one-way dark: $F_{3,8} = 8.64$, p = 0.007; Figure 5b). In contrast, the highest DOC effluxes in the light were at control temperatures, decreasing with both increasing and decreasing temperatures to DOC fluxes around zero (one-way: $F_{3,8} = 16.76$, p = 0.001; Figure 5b).

OA alone (at ambient temperatures) had a significant effect on DOC, shifting from a slight efflux at current-$p$CO$_2$ (~0.5 mmol-C m$^{-2}$ d$^{-1}$) to a significant uptake at future-$p$CO$_2$ (~10.9 mmol-C m$^{-2}$ d$^{-1}$; paired-sample: t = 5.74, df = 2, p = 0.03; Figure 5a). The trend in temperature response was similar for future and current-$p$CO$_2$ (two-way interaction: $F_{3,16} = 0.88$, p = 0.47; Figure 5a), but there was a shift from small efflux at lower temperatures to considerable uptakes at all temperatures (two-way CO$_2$ effect: $F_{1,16} = 61.46$, p <0.0001; Figure 5a). Differences between dark and light DOC fluxes were also independent of temperature (two-way interaction: $F_{3,16} = 1.94$, p = 0.16; Figure 5c) with the overall magnitude of fluxes in the dark being significantly greater than those in the light (two-way light-condition: $F_{1,16} = 15.83$, p = 0.001; Figure 5c). Loss of statistically





different temperature responses for light and dark responses (temperature effect two-way: $F_{3,16} = 1.05$, p = 0.40; Figure 5c)

was in large part due to within treatment variability in the future-$p$CO$_2$ cores.

## 4    Discussion

The aim of this study was to explore the changes in DOC demand and production in unvegetated estuarine sediments under a range of temperatures at current and future-$p$CO$_2$ levels. The purpose of this was to gain a better understanding of how unvegetated sediments contribute to estuarine DOC export and how this will change under projected future climate conditions.

An important component of the study was testing the interaction and individual effects of warming and OA on DOC processing. This was necessarily achieved through a comparison of core incubations occurring in different weeks. As such, it is important to consider the limitations of this approach. Control treatments in different weeks would ideally be the same in all respects, but there were some differences. For instance, NPP and R were higher in the incubation week for current-$p$CO$_2$ conditions (Table S2), likely due to small changes in environmental conditions, e.g. salinity differences (24 versus 17.7 for current and

future-$p$CO$_2$, respectively; Table 1). Yet, these differences did not significantly affect DOC fluxes, nor the heterotrophy of the sediments (P/R = 0.84 ± 0.01 and 0.83 ± 0.04; Table S2). Moreover, sediments in separate weeks maintained the same OM content (~3.5 %) and molar C:N ratio (~16), suggesting that differences in processing have very little short-term impact on the overall OM pool in the sediment due to the OM pool size being about 3 orders of magnitude higher than any diel flux (organic carbon pool ~12,000 mM). Thus, because all conditions in the laboratory setup were the same for each incubation (with the

exception of $p$CO$_2$ in treatment tanks, which was intentionally manipulated to be different) the difference in fluxes between controls were attributed to differences in when the sediments and overlying waters were collected. Therefore, the scaling of NPP and R (Table S3) were done for the sake of treatment comparison, resulting in scaled rates within 13% of actual measured values, which had a negligible effect on P/R (< 1 % across all treatments). The final NPP and R rates should thus be considered relative to control rates and be interpreted as approximate values (± 13 %).

Understanding current ecosystem functioning is of primary interest when trying to determine how disturbances in the environment may change metabolic rates and pathways of OM mineralization (Jørgensen, 1996; D'Avanzo et al., 1996; Malone and Conley, 1996). The near 1:1 ratio of DIC production to O$_2$ consumption in the dark (respiratory quotient, RQ of ~1.13 ± 0.05; Figure 4) suggests that aerobic respiration dominated the sediments (Eyre and Ferguson, 2002). Similarly, aerobic processes dominated the benthic production in the light as shown by the 1:1 ratio of O$_2$ and DIC fluxes (Fig. 4; Eyre and

Ferguson, 2002). Sediments here were net heterotrophic with a P/R in control cores of ~0.84 ± 0.01 and ~0.83 ± 0.04 during current and future-$p$CO$_2$ incubation weeks, respectively. Despite the undeniable range of P/R ratios unvegetated estuarine sediments may experience (1.2 to 0.01 in Oakes et al., 2012; and Ferguson and Eyre, 2013, respectively), the ratios here were similar to mean global model estimates for unvegetated estuarine sediments (~0.82, calculated from values in Duarte et al., 2005) and calculated from P an R values of 22 estuaries globally (~0.84, compiled by Smith and Hollibaugh, 1993), suggesting

that the metabolic function of sediments in the current study are representative of estuarine sediments globally and the impacts observed in this study should be broadly applicable.



### 4.1 DOC fuels benthic respiration

DOC appeared to be a significant driver of benthic respiration (Figure 5b). At control temperatures (23 °C) net DOC fluxes were near zero (0.47 ± 0.93 mmol-C m$^{-2}$ d$^{-1}$), indicating that the diel production and uptake of DOC across the sediment-water interface was balanced (Figure 5a). The control rates in the present study were close to benthic DOC flux rates reported for subtropical estuarine sediments in most seasons, ~1.5 mmol-C m$^{-2}$ d$^{-1}$, except summer (Maher and Eyre, 2010). Relative to our control (summer) rates, Maher and Eyre (2010) reported higher net DOC flux rates (~10 mmol-C m$^{-2}$ d$^{-1}$) as a result of DOC effluxes in both the light and dark (Maher and Eyre, 2010). Our light DOC fluxes (610 $\mu$mol-C m$^{-2}$ h$^{-1}$) were similar to those of Maher and Eyre (2010) in summer (~647 $\mu$mol-C m$^{-2}$ h$^{-1}$). The difference in the DOC processing in the sediments came from dark uptake (-571 $\mu$mol-C m$^{-2}$ h$^{-1}$) versus dark efflux in the previous study (254 $\mu$mol-C m$^{-2}$ h$^{-1}$, Maher and Eyre, 2010). This release of DOC in the dark was attributed to enhanced microbial coupling in the sediments under warmer temperatures (Maher and Eyre, 2010), yet here, and in previous reports, DOC uptake suggests that bacteria not only intercepted DOC produced from within the pore waters (potentially satisfying up to 60 % of total mean bacterial production, Boto et al., 1989), but also took up available DOC from the water column to satisfy its metabolic requirements (Boto et al., 1989; Brailsford et al., 2019), effectively acting as a DOC sink.

### 4.2 OA increases DOC uptake

Positive responses in primary production were associated with OA. The ~72 % increase in NPP rates at ambient temperatures were consistent with general stimulation of primary production in finer sediments with increased DIC availability (Vopel et al., 2018; Oakes and Eyre, 2014). Sediments may become DIC-limited when algal demand is relatively high compared to porewater supply of $CO_2$ (Cook and Røy, 2006), and MPB therefore may benefit from an increase in $CO_2$ availability. MPB in fine sediments are restricted to dissolved substrates (i.e., nutrients and DIC) accessed via diffusion from deeper and adjacent sediments, and the overlying water column (Boudreau and Jørgensen, 2001). This makes them more likely to deplete accessible DIC than MPB in permeable sediments. Primary producers in permeable sediments, like those in reef ecosystems, therefore do not often experience the same increase in primary production with increased $CO_2$ (Trnovsky et al., 2016; Cyronak and Eyre, 2016; Eyre et al., 2018; Cook and Røy, 2006; Vopel et al., 2018). As well as differences in  diffusive versus advective modes of solute transfer between the sediment types (Cook and Røy, 2006), but also may be partially due to sandier sediments being limited by other factors such as nutrient and OM availability as they are generally more oligotrophic (Admiraal, 1984; Heip et al., 1995). Therefore, DIC limitations to MPB growth rates are likely higher under low sediment permeability like those here and primary productivity responses would likely differ in permeable sediments where general access to $CO_2$ is greater.

Given that MPB exude carbon (Maher and Eyre, 2010), we would expect increased GPP to correspond with increased DOC production and flux. However, although OA stimulated primary production (Figure 2), we instead saw increased DOC uptake in the dark (Figure 5). A likely explanation is that bacterial uptake of DOC was stimulated through the provision of labile carbon from MPB (Morán et al., 2011; Hardison et al., 2013). As such, DOC appeared to fuel much of the dark cycle respiration, as DOC uptake in the dark reflected dark DIC production (respiration), except for sediments at Δ-3 °C under





current-$p$CO$_2$. Under current-$p$CO$_2$, the sediment uptake of DOC in the dark accounted for ~50 % of the total respired DIC.
This suggests that there was respiration of other carbon sources, potentially more refractory DOC sourced from within the pore
waters (Boto et al., 1989), with possibly more metabolic energy invested for the production of ectoenzymes needed to
hydrolyze this more refractory DOC (Chróst, 1990; Chróst, 2017). The portion of DIC accounted for by dark DOC increased
from 50 to 100 % under the future-$p$CO$_2$ climate. In part, this may be due to the increase in available labile organic carbon
(Moran and Hodson, 1990) arising from the increase in NPP under future-$p$CO$_2$ across all temperatures (Figure 2b). The
increase in the ratio of DOC uptake to DIC efflux from 0.5 to 1.0 may be due to the bacteria no longer needing to synthesise
ectoenzymes in the presence of readily utilizable organic carbon (Chróst, 1992; Chróst, 2017), resulting in a more rapid
turnover of carbon to the water column.

### 4.3    Warming drives increased heterotrophy and DOC assimilation increases

Sediments in this study, like other manipulative studies, in both permeable sands (Lantz et al., 2017; Trnovsky et al., 2016)
and cohesive sediments (Apple et al., 2006), demonstrated increased heterotrophy with increased temperature. This shift to
heterotrophy is often attributed to the imbalance of heterotrophic over autotrophic metabolic thermal sensitivity (Yang et al.,
2016; Allen et al., 2005). More specifically, differences in activation energy dictated by differences in physiology and
biochemical processes (Patching and Rose, 1970; Apple et al., 2006) result in greater increases in heterotrophic activity with
increasing temperature than autotrophic increases in activity (Yang et al., 2016). However, in this study, under current-$p$CO$_2$,
the increases in R and GPP from Δ-3 °C to control temperatures were similar (~16 % and ~11 %, respectively), whereas at
higher temperatures, GPP decreases far exceeded increases in R (7 times and 3 times, for 26 °C and 28 °C respectively).
Therefore, unlike previous studies, decreases in MPB productivity at higher temperatures appeared to be a greater driver
towards heterotrophy than increases in respiration rates. In other words, temperature increases not only increased the rate of
DOC uptake, but also likely decreased the rate of DOC production.

### 4.3.1    Warming reduces GPP and DOC production under current-$p$CO$_2$

Primary production is the main source of DOC in marine ecosystems (Wagner et al., 2020). Decreasing trends in GPP with
warming under current-$p$CO$_2$ seen here have been described previously where photosynthetic growth and production decline
at higher temperatures (Thomas et al., 2012). Photosynthetic productivity is often linked to seasonal temperature (Apple et al.,
2006), which is also associated with differences in environmental factors such as light, nutrient concentrations, and DOM
quality and availability (Geider, 1987; Herrig and Falkowski, 1989). Although the relative availability of light and nutrients
do influence productivity rates (Kana et al., 1997) and would be expected to influence in situ seasonal production, the current
study controlled light and initial nutrient concentrations in the water column to isolate the effect of temperature. Thus,
decreasing GPP was driven by warming, suggesting that MPB in these subtropical sediments had a temperature optimum
around current mean summer temperatures of ~23 °C (GPP: $1515 \pm 37$ $\mu$mol-O$_2$ m$^{-2}$ h$^{-1}$; Figure 2c).





### 4.3.2 Warming increases respiration and DOC uptake

Unlike photosynthetic productivity, heterotrophic respiration often has a linear rate increase with temperature to the thermal optimum due to heterotrophs not being constrained by the same abiotic variables (e.g., nutrient and light availability) as primary production (Apple et al., 2008; Apple et al., 2006; Geider, 1987; Yap et al., 1994). In this study, respiration rates under both

current and future-$p$CO$_2$, increased from lowest rates measured at $\Delta$-3 °C to maximum rates (>50 % greater) at $\Delta$+5 °C (Figure 2a). Consistent with overall lower respiration rates relative to other subtropical unvegetated sediments (~900 to ~1500 $\mu$mol-O$_2$ m$^{-2}$ h$^{-1}$, Ferguson and Eyre, 2013) the temperature dependence of respiration under both current and future-$p$CO$_2$ conditions ($Q_{10}$ = 1.66 and 1.69, respectively) was slightly lower than is typical for biological systems ($Q_{10}$ = 2, Valiela, 1995), but similar to temperature dependence described in other estuarine systems ($Q_{10}$ = 1.5-1.9, Morán et al., 2011), with values towards the

lower end of this range possibly being a result of resource limitation (López-Urrutia and Morán, 2007).

A potential limiting resource for bacteria in estuarine sediments is dissolved organic matter (DOM) (Church, 2008), ultimately controlling the flow of carbon through the microbial loop (Kirchman and Rich, 1997). However, in the presence of sufficient DOM, warming has been associated with increased bacterial DOM incorporation (Kirchman and Rich, 1997). In line with this, an increased uptake of DOC at higher temperatures and efflux at lower temperatures was observed. Although DOC is mainly

produced by photoautotrophs, DOC can be produced in the dark (i.e., cell lysis via viruses and potential bacterial grazing via meograzer, Carlson, 2002). As such, the efflux of DOC in the dark at $\Delta$-3 °C suggests that heterotrophic bacterial productivity, and therefore DOC uptake, was reduced by lowered temperatures (Raymond and Bauer, 2000), resulting in a failure to intercept all DOC produced in the pore waters. This failure to intercept DOC may be compounded if nutrient supply is limited (Brailsford et al., 2019), as it is common for heterotrophic bacteria to rely on refractory DOC under such conditions (Chróst, 2017).

### 4.3.3 Global estuarine loss of DOC from unvegetated sediments in the future

Up to 3.5 times more DOC reaches the ocean interior from coastal areas than the open ocean (Costanza et al., 1997). As such, small changes to the coastal export of DOC may have a disproportionately large influence on the global DOC budget. Our findings suggest a reduced export of DOC to the ocean under future-$p$CO$_2$ conditions, across the full 8 °C temperature range in unvegetated sediments. Despite the lack of seasonality in the study, the inclusion of an 8 °C temperature range, including

temperatures below current mean temperatures, suggests that seasonal temperature variation is unlikely to have a significant effect on the relative change in DOC in the future (Figure 5). Although any upscaling of a single controlled experiment to a global scale is highly speculative, we feel it is better to include an estimate to demonstrate the potential changes that may transpire under a future high-$p$CO$_2$ climate and the potential importance of unvegetated sediments in DOC export from coastal zones by putting our findings in a global context, than not to attempt an estimate at all. The following estimates should be

considered in this context. Moreover, we have applied our results to global coastal DOC exports (Maher and Eyre, 2010; Duarte, 2017) as an initial step in modelling responses of unvegetated sediment habitats to future high-$p$CO$_2$ climate. We do not assume that the responses of unvegetated sediments to the future climate found here are applicable to other ecosystems dominated by macrophytes, and thus did not apply our findings to vegetated coastal habitats.





To estimate total DOC export from coastal zone under a future-$p$CO$_2$ climate of $\Delta$+3 °C and OA, the sediment uptake rate of

19 ± 4 mmol-C m$^{-2}$ d$^{-1}$ was scaled to the global surface area of unvegetated estuarine sediments (1.8 x 10$^{12}$ m$^2$; Costanza et al., 1997). An estimated 150 Tg-C would be removed from the coastal zone by unvegetated estuarine sediments annually. To then calculate the potential impact of this uptake, we applied our estimates to existing future global coastal DOC export estimates (Maher and Eyre, 2010; Duarte, 2017). Mean benthic DOC export from estuaries, including intertidal and vegetated habitats, has been estimated at 168 Tg-C yr$^{-1}$ (90-247 Tg-C yr$^{-1}$) (Maher and Eyre, 2010). Under this scenario, the switch to DOC uptake

by sediments under future climate conditions (Figure 5a) would result in an ~90 % reduction in total mean benthic estuarine DOC export (Maher and Eyre, 2010), decreasing the load from ~168 Tg-C yr$^{-1}$ to ~18 Tg-C yr$^{-1}$. Other global estimates of DOC exported from coastal vegetated ecosystems range from 114 up to 1,853 Tg-C yr$^{-1}$ (Duarte, 2017), with model estimates suggesting unvegetated estuarine sediments may consume 8 to 132 % of this DOC under a future-$p$CO$_2$ climate. As such, this basic modelling suggests that by impacting DOC fluxes in unvegetated sediments, future climate conditions could significantly

impact global DOC export from coastal systems to the open ocean. This has implications for global marine productivity and carbon transfer to the ocean interior (Krause-Jensen and Duarte, 2016). However, to get a more accurate insight into global carbon cycling the response of DOC export from estuarine vegetated habitats to future climate also needs to be studied.

**Team list**

Michelle Simone, Kai Schulz, Joanne Oakes, Bradley Eyre

**Data availability**

Archived data will be available for access on PANGAEA upon publication (currently under review). Until then, data available upon request.

**Author contributions**

All listed authors have contributed substantially to preparation and drafting of this paper and have approved the final submitted manuscript. Specifically, MS conceived the project, collected data, ran data analysis and interpretation, and led the writing of the manuscript. KS, JO, and BE helped conceive the project, contributed to interpretation and helped draft the manuscript.

**Competing interests**

The authors declare that they have no conflict of interest


**Acknowledgements**

Thanks are extended to P. Kelly, I. Alexander, M. Carvalho, N. Carlson-Perret, J. Yeo, and N. Camillini for their assistance in the field and support in the laboratory. Special thanks to Z. Kennedy allowing access to their property for sample collection. This work was supported in the form of an SESE Postgraduate Scholarship from Southern Cross University, Lismore, NSW,

Australia and ARC Discovery projects DP150102092 and DP160100248.




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





**Table 1. Start conditions for current and future $p$CO$_2$ (*) incubations showing mean (±SD) of various carbonate parameters. CON\* is the overlapping control core present in the future-$p$CO$_2$ incubation week.**

| Scenario | Sal (‰) | T (ºC) | pH (Free Scale) | $p$CO$_2$ ($\mu$atm) | HCO$_3^-$ ($\mu$mol/ kgSW) | CO$_3^{2-}$ ($\mu$mol/ kgSW) | TA ($\mu$mol/ kgSW) | DIC ($\mu$mol/ kgSW) |
|---|---|---|---|---|---|---|---|---|
| **Δ-3 °C** | 24.4 | 21.0 | 8.08 | 453.1 | 1750.8 | 123.9 | 2048.7 | 1889.8 |
|  |  | (±0.1) | (±0.02) | (±24.0) | (±2.9) | (±6.4) | (±12.6) | (±3.3) |
| * | 17.7 | 20.8 | 7.60 | 989.8 | 1232.9 | 24.28 | 1293.8 | 1291.6 |
|  |  | (±0.1) | (±0.02) | (±40.7) | (±2.6) | (±0.9) | (±2.8) | (±2.9) |
| **Control** | 24.4 | 23.1 | 8.07 | 469.9 | 1744.7 | 130.2 | 2056.9 | 1889.6 |
|  |  | (±0.0) | (±0.02) | (±27.2) | (±4.1) | (±6.9) | (±12.1) | (±2.0) |
| * | 17.7 | 23.2 | 7.63 | 995.9 | 1281.6 | 29.5 | 1354.7 | 1343.4 |
|  |  | (±0.1) | (±0.06) | (±146.6) | (±5.5) | (±4.3) | (±12.7) | (±6.1) |
| **Δ+3 °C** | 24.4 | 25.6 | 8.08 | 471.5 | 1723.5 | 136.9 | 2051.6 | 1874.5 |
|  |  | (±0.5) | (±0.01) | (±13.3) | (±2.0) | (±3.4) | (±6.2) | (±1.5) |
| * | 17.7 | 25.8 | 7.64 | 1011.1 | 1265.8 | 32.7 | 1346.7 | 1329.2 |
|  |  | (±0.2) | (±0.12) | (±248.6) | (±2.7) | (±9.3) | (±23.6) | (±3.4) |
| **Δ+5 °C** | 24.4 | 27.1 | 8.11 | 445.2 | 1698.3 | 155.3 | 2069.3 | 1866.1 |
|  |  | (±0.1) | (±0.05) | (±56.2) | (±22.7) | (±17.0) | (±17.6) | (±7.4) |
| * | 17.7 | 27.9 | 7.65 | 989.6 | 1254.4 | 34.3 | 1339.2 | 1317.1 |
|  |  | (±0.1) | (±0.12) | (±40.7) | (±5.2) | (±1.3) | (±3.8) | (±5.2) |
| **CON\*** | **17.7** | **23.3** | **7.96** | **431.9** | **1193.0** | **58.4** | **1338.2** | **1265.5** |
|  |  | **(±0.1)** | **(±0.05)** | **(±45.7)** | **(±4.1)** | **(±6.1)** | **(±10.9)** | **(±1.1)** |

**Table 2. Q$_{10}$ and T$_{opt}$ values for current and future-$p$CO$_2$ climates**

|  | R | | NPP | | GPP | |
|---|---|---|---|---|---|---|
|  | Current | Future | Current | Future | Current | Future |
| Q$_{10}$ | 1.66 | 1.69 | 1.13 | 1.92 | 1.46 | 2.27 |
| T$_{opt}$ (°C) | 28 | 28 | 23 | 23 | 23 | 23 |


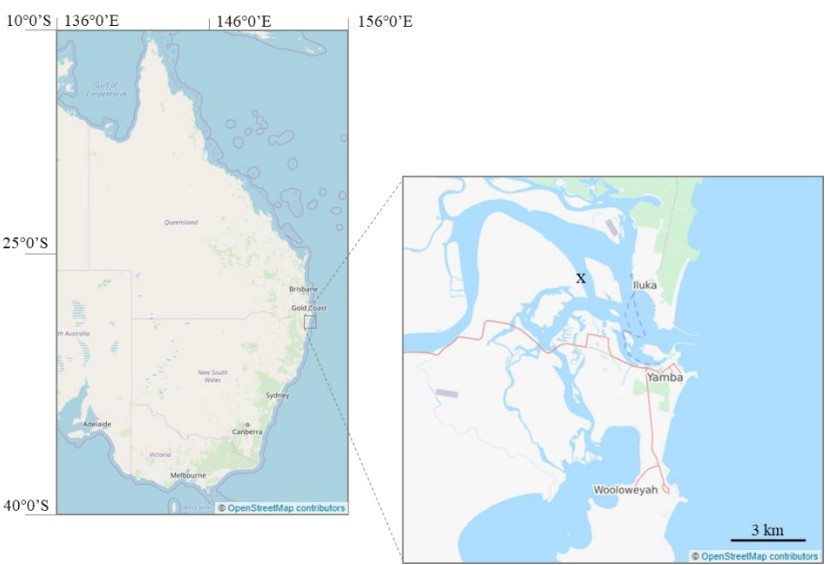

**Figure 1. Study location (x; 29°24.21'S, 153°19.44'E) marked on a map of Yamba, NSW embedded in an east coast map of Australia. © OpenStreetMap contributors 2020. Distributed under a Creative Commons BY-SA License.**

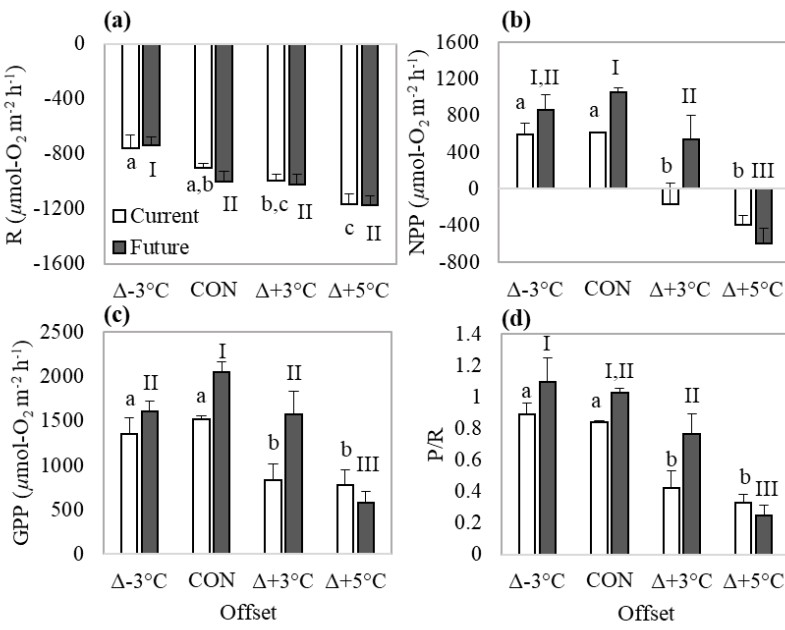

**Figure 2. Effect of temperature (a) R rates (b) NPP rates, (c) GPP rates ($\mu$mol-$O_2$ m$^{-2}$ h$^{-1}$) and (d) P/R under current and future-**
**$p$CO$_2$. Panels show mean (± SD) values at three temperature offsets from control (CON), 23 °C. Letters identify significantly different means across temperatures under current-$p$CO$_2$ and numerals identify significantly different means across temperatures under future-$p$CO$_2$ conditions. Letters or numerals that are the same indicate no significant difference.**

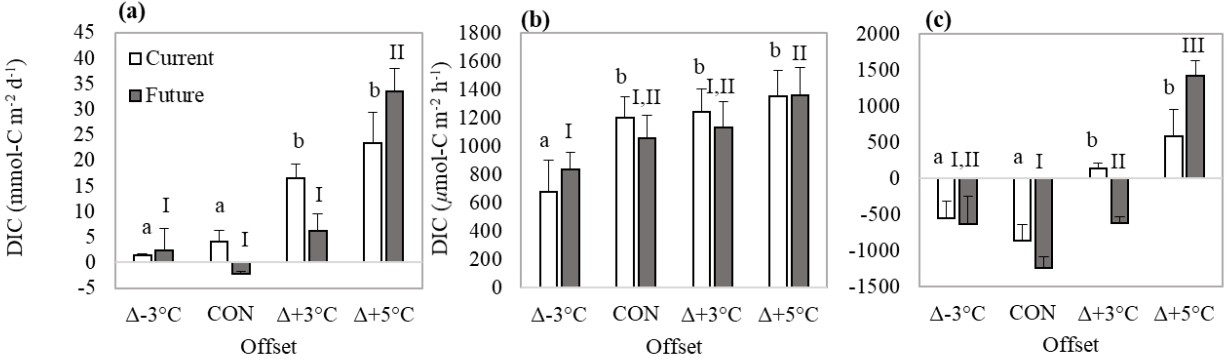

**Figure 3. Effect of temperature on DIC fluxes in the (a) net DIC production (mmol-C m$^{-2}$ d$^{-1}$), and (b) dark and (c) light ($\mu$mol-C m$^{-2}$ h$^{-1}$) under current and future-$p$CO$_2$. Panels show mean (± SD) rates at three temperature offsets from control (CON), 23 °C. Letters identify significantly different means across temperatures under current-$p$CO$_2$ and numerals identify significantly different means across temperatures under future-$p$CO$_2$ conditions. Letters or numerals that are the same indicate no significant difference.**

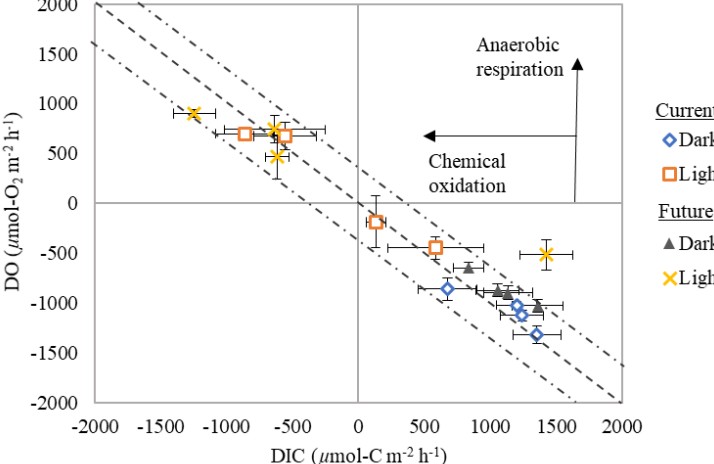

**Figure 4. DIC:DO fluxes from sediment ($\mu$mol-C or -O$_2$ m$^{-2}$ h$^{-1}$) for all temperatures in dark and light cycles subject to current and future-$p$CO$_2$ (mean ± SD) Dashed line highlights the 1:1 ratio (± 18%, Hopkinson, 1985) with values falling on this line likely a result of aerobic respiration. Arrows indicate the position values would fall in if sediments were experiencing chemical oxidation or anaerobic respiration.**





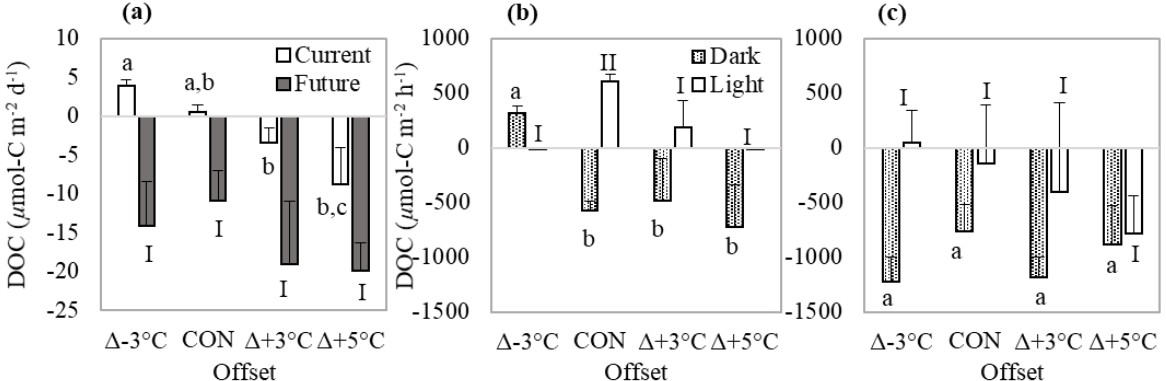


**Figure 5. Effect of temperature on DOC fluxes under current and future-$pCO_2$. Panel (a) shows mean (± SD) net DOC fluxes ($\mu$mol-C m$^{-2}$ d$^{-1}$) under current and future-$pCO_2$ conditions at three temperature offsets from control (CON), 23°C. Letters identify significantly different means across temperatures under current-$pCO_2$ and numerals identify significantly different means across temperatures under future-$pCO_2$ conditions, where letters or numerals that are the same indicate no significant difference.**

**Light and dark fluxes for DOC ($\mu$mol-C m$^{-2}$ h$^{-1}$) are presented in Panels (b) and (c) for current-$pCO_2$ and future-$pCO_2$ conditions, respectively. Here, letters identify significantly different means across temperatures in dark and numerals identify significantly different means across temperatures in light cycles. Letters or numerals that are the same indicate no significant difference.**

Supplementary tables available in "Supplement"

**Table S1. Measured total alkalinity (TA) and DIC used to calculate pH (Free scale) using CO$_2$SYS directly compared to the measured**
**pH from the cores using HACH multiprobe meter with pH probe. Mean absolute difference was used to estimate uncertainty in $pCO_2$ calculations via CO$_2$SYS. Data used in a manuscript currently under review.**

**Table S2. Overlapping mean control rates (±SD) in current and future-$pCO_2$ incubations for dark and light cycles. Units for dark and light rates ($\mu$mol-C or -O$_2$ m$^{-2}$ h$^{-1}$) and net rates (mmol-C or -O$_2$ m$^{-2}$ d$^{-1}$). Scaled means in Table S3 applied to significantly**
**different means (*) only.**

**Table S3. Scaled means (± SD) for R and NPP rates ($\mu$mol-O$_2$ m$^{-2}$ h$^{-1}$) under current and future-$pCO_2$ incubations. CON* is the overlapping control present both weeks (note: current control and CON current are the same).**