# Peer review of "Warming and ocean acidification may decrease estuarine dissolved organic carbon export to the ocean"

_Biogeosciences, 2020_

## Referee Comment (RC1) · Anonymous Referee #1 · 23 Nov 2020

Review of bg-2020-335 Warming and ocean acidification may decrease estuarine dissolved organic carbon export to the ocean Michelle N. Simone, Kai G. Schulz, Joanne M. Oakes, and Bradley D. Eyre

This contribution studies the effect of increased pCO2 and temperature on the fate of DOC in photic sediments. There are two autochthonous sources for DOC in sediments: degradation of detrital POC and release from microphytobenthos. Diffusive fluxes between the overlying water and sediment pore water depend on the concentration gradient (excluding bioturbation in more permeable sediments). Increases in pCO2 will be expected to enhance benthic primary production (and associated DOC production) while increases in temperatures will increase carbon mineralisation rates. The net effect of these combined is difficult to assess and hence the focus of this experimental study. The experiment is very well designed and carried out, and the results are clearly condensed and presented. The results and discussion sections are, however, difficult reading, and I had to re-read many times to follow. I wonder if the carbon budget/fluxes can be summarised in a figure or table so it is easier for the reader to follow the net result of the treatments. I found myself doing this while reading the discussion, gathering numbers from different figures. This would great increase the impact of the paper. I found it misleading to always refer to the high pCO2 scenario as ocean acidification OA. It is the increased DIC availability that is fuelling higher primary production which seems to be the major driver, rather than acidification influencing a rate as such. I recommend that this is rectified. It is also unclear what the nutrient levels were during the experiment. The results and discussion are focused solely on carbon limitation and assume adequate nutrient supply. That said the system the sediment cores were sampled from appears to be low nutrient. It is worth addressing this at some point. What effect would N limitation have on the result. Competition between MPB and heterotrophs for available nutrients for example. Finally, I do not see the value in scaling the data up to global estimates of sediment estuarine DOC uptake (4.3.3). It is not necessary and is fraught with very large assumptions. Similar scale ups have been done in the cited literature (Duarte papers), arrive at questionable results and conflict with current understanding of the global ocean DOC budget. The findings of this present study are relevant, intriguing and warrant publication without this final section.

Specific comments Introduction Important to distinguish between photic and aphotic sediments. They differ greatly in their role and contribution to the larger net effects of coastal waters, which are outlined at the start of the introduction. The last part of the introduction could be rephrased to be clearer. Lines 54-78. First formulate what the dominating mechanisms acting on DOC uptake/release from photic sediments are. Then address how these mechanisms can be influenced by warmer temperatures, high CO2, and lowered pH, respectively. Then clearly state the hypothesis you had as the

basis of your experimental design.

What influence would variable light conditions have on your findings? The cores are taken from a shallow estuarine site where one can expect considerable resuspension from tides, currents and winds. The light intensities used here are likely representative of best case. So one can maybe amplify the dark scenario?

Line 7. " Estuaries make a disproportionately". What do you mean here? With respect to what?

Line 19. DOC is smaller than that retained in soils and also in fossil fuels.

Line 28. And line 32-35. Here you state that 33% of the NPP in coastal waters is exported to the oceans and stored in the ocean interior. I question the validity of this statement/citation. Is there evidence that the interior ocean is increasing in DOC? Why the large difference between mineralisation efficiency of DOC produced in surface water of the ocean to that produced in coastal waters?

Line 43. Delete extra "lability"

First three paragraphs contradict. You start by arguing that coastal waters are an important source of DOC to the open ocean but then finish by stating that coastal sediments are an important sink for DOC.

Line 48. Check referencing. Fischot and Benner paper does not address the processing of DOC by estuarine sediments.

Line 55-60. The increased DOC production in the Engel et al 2013 study was due to nutrient limitation. When they added nutrients it was rapidly removed again. So no net accumulation of DOC.

Line 287-289. This can be deleted.

Line 340-343. Check phrasing and possible break into two sentences to make easier reading.

Line 350-359. Here the authors begin to speculate about the lability of DOC without any measurements to support it. I am not sure it is necessary.

Line 395. DOC is also produced continually from the detrital sediment POC. This contributes to dark DOC production.

Lin 398-399. Are you inferring nutrient limitation in your set up? For now I have assumed you had adequate nutrients.

Line 401. A very bold statement and the reference (Costanza) does not seem to support it. Please check.

Figures Error bars in the figure should go both plus and minus. Check text in figure 4. Do you not mean aerobic respiration (with arrow pointing upwards).

---

## Referee Comment (RC2) · Anonymous Referee #2 · 27 Nov 2020

This is a well described experimental case study that contributes to close an important knowledge gap concerning the modification of the carbon cycle under global environmental and climatic change. My biggest concern in the study is the upscaling to the global dimension. The authors are aware of the associated risks and that such an upscaling may be (at least) quantitatively quite problematic. Overall, this is a thoroughly made study and a useful addition in the field.

Suggestions for a revised manuscript:

Section 4.3.3: The authors are correct in being very careful when they provide a daring global upscaling here. It would be good to add a paragraph on detailing why such an

upscaling can be risky and possibly incorrect (different hydrodynamic settings, different sediment composition, different delivery of dissolved and particulate matter from land and through aeolian deposition, etc.)

Line 54: It is not only the climate project models but rather the scenarios used for the projections. The scenarios are usually produced through simplified climate models and integrated assessment models.

Line 55: "increasing the partial pressure by 580 ppm" – relative to which reference year?

Lines 55-60: Though regionally primary production may be enhanced with temperature and pCO2, climate change can lead to increased stratification and a decrease of mixing as well. It would be good to also discuss this aspect and cite a few relevant literature sources.

Line 140: "refit from Mehrbach et al. (1973)" – can you describe in more detail how and why you did this?

Line 277: "OA alone (at ambient temperatures)" – what is meant with 'ambient temperatures' exactly?

Section headings "4.2 OA increases DOC uptake" and "4.3.2 Warming increases respiration and DOC uptake" are unclear. Which component takes up DOC? Maybe use a different word for 'uptake'?

Figure 1: Some fonts are so tiny that they are not readable. Please, increase them if relevant or delete unnecessary information.

Figure 5: The 'bars' within the grey and dotted areas of the plot are barely visible. What do these 'bars' show? Please, provide information in the figure caption.

---

## Author Comment (AC2) · 16 Dec 2020

**Review of bg-2020-335 Warming and ocean acidification may decrease estuarine dissolved organic carbon export to the ocean Michelle N. Simone, Kai G. Schulz, Joanne M. Oakes, and Bradley D. Eyre**

"This is a well described experimental case study that contributes to close an important knowledge gap concerning the modification of the carbon cycle under global environmental and climatic change. My biggest concern in the study is the upscaling to the global dimension. The authors are aware of the associated risks and that such an upscaling may be (at least) quantitatively quite problematic. Overall, this is a thoroughly made study and a useful addition in the field.

Suggestions for a revised manuscript:"

**Comment:** Section 4.3.3: The authors are correct in being very careful when they provide a daring global upscaling here. It would be good to add a paragraph on detailing why such an upscaling can be risky and possibly incorrect (different hydrodynamic settings, different sediment composition, different delivery of dissolved and particulate matter from land and through aeolian deposition, etc.)

**Reply:** We agree. This section is highly speculative and is purely an exercise of interest, a likely exercise that readers will do on their own. We will follow Reviewer 2's suggestion and add further details regarding the limitations of the upscaling. Also, see our reply to Reviewer 1's comments.

**Comment:** Line 54: It is not only the climate project models but rather the scenarios used for the projections. The scenarios are usually produced through simplified climate models and integrated assessment models.

**Reply:** Yes, this is true. We had included the scenario reference at the end of the sentence (RCP8.5), however, it would be more forthcoming to include the "high-emission scenario climate projections" explicitly in the text. This adjustment will be added. "Climate projection models under a high-emission scenario suggest that atmospheric $CO_2$ concentrations could more than double by the end of the century, increasing the partial pressure of $CO_2$ ($pCO_2$) in surface waters to 1000 $\mu$atm and decreasing pH by 0.3 units, together termed ocean acidification (OA) (RCP8.5, IPCC, 2019)."

**Comment:** Line 55: "increasing the partial pressure by 580 ppm" – relative to which reference year?

**Reply:** This has been rewritten for clarity. LN 54: "Climate projection models suggest that atmospheric $CO_2$ concentrations could more than double by the end of the century, increasing the partial pressure of $CO_2$ ($pCO_2$) in surface waters to 1000 $\mu$atm …"

**Comment:** Lines 55-60: Though regional primary production may be enhanced with temperature and $pCO_2$, climate change can lead to increased stratification and a decrease of mixing as well. It would be good to also discuss this aspect and cite a few relevant literature sources.

**Reply:** This discussion of the possible effect of stratification will be added to the discussion section with the following text in section4.3 (Warming drives increased heterotrophy and DOC assimilation): "Although it is yet to be assessed directly, the enhancement of primary production from temperature and $p$CO$_2$ may be counteracted by a potential increase in stratification with the changing climate (Li et al., 2020). Increased stratification has the potential to decrease nutrient supply to primary producers, despite increased light availability (Rost et al., 2008). However, this stratification is more likely to impact phytoplankton in oligotrophic waters rather than benthic microalgae which are in direct contact with remineralised nutrient supplied from within sediments."

**Comment**: Line 140: "refit from Mehrbach et al. (1973)" – can you describe in more detail how and why you did this?

**Reply:** We did not do the refit, Dickson and Millero (1987) did. The sentence reads, "Total borate concentrations (Uppström, 1974) and boric acid (Dickson, 1990) and stoichiometric equilibrium constants for carbonic acid (Dickson and Millero, 1987), refit from Mehrbach et al. (1973), were used." We just wanted to include the original source of Dickson and Millero (1987). For clarity, this has been rewritten as "…carbonic acid from Mehrbach et al. (1973) as refit by Dickson and Millero (1987), were used."

**Comment:** Line 277: "OA alone (at ambient temperatures)" – what is meant with 'ambient temperatures' exactly?

**Reply:** At ambient temperatures was meant to distinguish the OA scenario from the OA and temperature manipulation scenarios. This would therefore be at 23 °C. This sentence would be improved with the addition of the temperature included. LN 277 will now read, "OA alone (at mean ambient temperatures, 23 °C)"

**Comment:** Section headings "4.2 OA increases DOC uptake" and "4.3.2 Warming increases respiration and DOC uptake" are unclear. Which component takes up DOC? Maybe use a different word for 'uptake'?

**Reply:** We can see the ambiguity in uptake. We believe assimilation would be a more accurate term as the heterotrophs in the sediments actively assimilate DOC. The section headings will now read: "4.2 OA increases DOC assimilation" and "4.3 Warming drives increased heterotrophy and DOC assimilation"

**Comment:** Figure 1: Some fonts are so tiny that they are not readable. Please, increase them if relevant or delete unnecessary information.

**Reply:** This will be adjusted as suggested.

**Comment:** Figure 5: The 'bars' within the grey and dotted areas of the plot are barely visible. What do these 'bars' show? Please, provide information in the figure caption.

**Reply:** The figure has been redesigned. The figure caption will clearly indicate "Light (grey boxes) and dark fluxes (spotted boxes) of DOC ($\mu$mol-C m$^{-2}$ h$^{-1}$) for (b) current-$p$CO$_2$ and (c) high-$p$CO$_2$ conditions."

**References**

Dickson, A. G., and Millero, F. J.: A comparison of the equilibrium constants for the dissociation of carbonic acid in seawater media, Deep Sea Research Part A. Oceanographic Research Papers, 34, 1733-1743, doi: 10.1016/0198-0149(87)90021-5, 1987.

Dickson, A. G.: Thermodynamics of the dissociation of boric acid in potassium chloride solutions from 273.15 to 318.15 K, J. Chem. Eng. Data., 35, 253-257, doi: 10.1021/je00061a009, 1990.

IPCC: Special Report on the Ocean and Cryosphere in a Changing Climate, 2019.

Li, G., Cheng, L., Zhu, J., Trenberth, K. E., Mann, M. E., and Abraham, J. P.: Increasing ocean stratification over the past half-century, Nature Climate Change, 10, 1116-1123, doi: 10.1038/s41558-020-00918-2, 2020.

Mehrbach, C., Culberson, C. H., Hawley, J. E., and Pytkowicx, R. M.: Measurement of the Apparent Dissociation Constants of Carbonic Acid in Seawater at Atmospheric Pressure, Limnol. Oceanogr., 18, 897-907, doi: 10.4319/lo.1973.18.6.0897, 1973.

Rost, B., Zondervan, I., and Wolf-Gladrow, D.: Sensitivity of phytoplankton to future changes in ocean carbonate chemistry: Current knowledge, contradictions and research directions, Marine Ecology-progress Series - MAR ECOL-PROGR SER, 373, 227-237, doi: 10.3354/meps07776, 2008.

Uppström, L. R.: The boron/chlorinity ratio of deep-sea water from the Pacific Ocean, Deep Sea Research and Oceanographic Abstracts, 1974, 161-162,

---

## Author Response (AR1)

**Review of bg-2020-335 Warming and ocean acidification may decrease estuarine dissolved organic carbon export to the ocean Michelle N. Simone, Kai G. Schulz, Joanne M. Oakes, and Bradley D. Eyre**

This contribution studies the effect of increased $pCO_2$ and temperature on the fate of DOC in photic sediments. There are two autochthonous sources for DOC in sediments: degradation of detrital POC and release from microphytobenthos. Diffusive fluxes between the overlying water and sediment pore water depend on the concentration gradient (excluding bioturbation in more permeable sediments). Increases in $pCO_2$ will be expected to enhance benthic primary production (and associated DOC production) while increases in temperatures will increase carbon mineralisation rates. The net effect of these combined is difficult to assess and hence the focus of this experimental study. The experiment is very well designed and carried out, and the results are clearly condensed and presented.

**Comment:** The results and discussion sections are, however, difficult reading, and I had to re-read many times to follow.

**Reply:** In addition to addressing the specific comments of both reviewers, the results and discussion section will be revised to improve clarity and readability.

Reply: Line numbers have been adjusted throughout this document to reference the revised text in response to this and the rest of the reviewer comments (see below).

**Comment:** I wonder if the carbon budget/fluxes can be summarised in a figure or table so it is easier for the reader to follow the net result of the treatments. I found myself doing this while reading the discussion, gathering numbers from different figures. This would great increase the impact of the paper.

**Reply:** Figures 2, 3 and 5 already provide a summary of flux data referred to in the main text. However, we appreciate that some readers may find it easier to refer to a table. We are happy to build a summary table of fluxes. To avoid duplication this will be included as an appendix to the manuscript.

Reply: In the supplementary material you will now find Table S4-S6 with data requested by Reviewer 1. Captions read as follows:

Table S1. Gross primary productivity (GPP) and productivity to respiration ratio (P/R) calculated for each temperature under both current and high-$pCO_2$.

Table S2. Dark and light fluxes of dissolved organic carbon (DOC) for each temperature under both current and high-$pCO_2$.

Table S3. Dark and light fluxes of dissolved inorganic carbon (DIC) for each temperature under both current and high-$pCO_2$.

**Comment:** I found it misleading to always refer to the high $pCO_2$ scenario as ocean acidification OA. It is the increased DIC availability that is fuelling higher primary production which seems to be the major driver, rather than acidification influencing a rate as such. I recommend that this is rectified.

**Reply:** We agree with the reviewer, the use of OA and high-$pCO_2$ will be simplified as a reference to high-$pCO_2$ only.

35 Reply: In the introduction we found it necessary to keep the use of OA for context, however, at the end of the introduction we have added text to highlight the distinction between OA and high-$pCO_2$. The text now reads:

LN 85: Moreover, despite the potential stimulation of primary productivity in unvegetated muddy sediments by OA (Vopel et al., 2018) or more likely high-$pCO_2$, and potential enhancement of DOC production (Engel et al., 2013; Liu et al., 2017), this increase in labile DOC may promote bacterial productivity and DOC mineralisation (Hardison et al., 2013).

40 **Comment:** It is also unclear what the nutrient levels were during the experiment. The results and discussion are focused solely on carbon limitation and assume adequate nutrient supply. That said the system the sediment cores were sampled from appears to be low nutrient. It is worth addressing this at some point.

**Reply:** Nutrients did not appear to be limiting in any of the treatments as nutrient concentration increased during all incubations. This will be outlined in the text and the methods and data in the table below will be included in supplementary

45 information.

In the text:

LN 353: "In comparison, nutrients were non-limiting in the less permeable sediments used in the current study, based on nutrient concentrations that increased during all incubations (see supplementary methods and Table S7)."

Supplementary methods:

50 Dissolved inorganic nitrogen (DIN) samples were collected at the start and end of the flux incubations and syringe-filtered (0.45 $\mu$m cellulose acetate) into duplicate 10 mL polyethylene vials with a headspace, and stored frozen. Samples were analysed colorimetrically using a Lachat$^{TM}$ flow-injection system as described in Eyre and Pont (2003).

**Table S7. DIN concentrations ($\mu$M) (mean ± standard deviation) at the start (minimum) and end of the full incubation cycle.**

| Treatment | Current-$pCO_2$ | | High-$pCO_2$ | |
| --- | --- | --- | --- | --- |
| | Start | End | Start | End |
| Δ-3 | 1.19 | 2.02 | 1.85 | 6.66 |
| | (± 0.01) | (± 0.45) | (± 0.27) | (± 1.36) |
| Control | 1.85 | 4.00 | 2.42 | 6.11 |
| | (± 0.16) | (± 0.27) | (± 1.01) | (± 1.39) |
| Δ+3 | 1.88 | 4.47 | 1.97 | 9.61 |
| | (± 0.42) | (± 2.10) | (± 0.31) | (± 1.36) |
| Δ+5 | 2.37 | 15.52 | 2.40 | 14.68 |
| | (± 0.18) | (± 1.81) | (± 0.58) | (± 4.42) |

55 **Comment:** What effect would N limitation have on the result. Competition between MPB and heterotrophs for available nutrients for example.

**Reply:** This comment from Reviewer 1 addresses an important possibility in the system. We have discussed the potential effect of nutrient limitation on DOC flux in LN 412: "This failure to intercept DOC may be compounded if nutrient supply is

limited (Brailsford et al., 2019), as it is common for heterotrophic bacteria to rely on refractory DOC when labile sources are
60 not readily available (Chróst, 1991), which can occur under conditions of nutrient limited biological productivity (Allen,
1978)." (with underlined sections adjusted for clarity)

**Comment:** Finally, I do not see the value in scaling the data up to global estimates of sediment estuarine DOC uptake (4.3.3).
It is not necessary and is fraught with very large assumptions. Similar scale ups have been done in the cited literature (Duarte
papers), arrive at questionable results and conflict with current understanding of the global ocean DOC budget. The findings
65 of this present study are relevant, intriguing and warrant publication without this final section.

**Reply:** We thank the reviewer for their positive comments on the relevance and interest of this study. We do, however,
acknowledge the limitations of the upscaling included in the manuscript. This exercise was intended to provide a more
qualitative perspective on the potential impact a future high-$p$CO$_2$ climate could have on the DOC export from estuaries. We
believe it is interesting to consider the role of unvegetated sediments in an ecosystem/global context as this system is often
70 overlooked in carbon budgets, whereas our upscaling exercise highlights the potential importance of processes (and changes
to those processes) in this environment. To address the concerns of reviewer 1, and as per Reviewer 2's suggestion, we will
add additional details of why such upscaling can be risky and possibly incorrect, including limitations such as different
hydrodynamic settings, different sediment composition, different delivery of dissolved and particulate matter from land and
through aeolian deposition, etc.

75 **Comment:** Specific comments Introduction **(1)** Important to distinguish between photic and aphotic sediments. They differ
greatly in their role and contribution to the larger net effects of coastal waters, which are outlined at the start of the introduction.
**(2)** The last part of the introduction could be rephrased to be clearer. Lines 54-78. First formulate what the dominating
mechanisms acting on DOC uptake/release from photic sediments are. Then address how these mechanisms can be influenced
by warmer temperatures, high CO$_2$, and lowered pH, respectively. Then clearly state the hypothesis you had as the basis of
80 your experimental design.

**Reply: (1)** We agree. To clarify our focus on euphotic sediments – the restatement of this focus will be added to the final
paragraph. LN 82: "We expected that warming would promote a stronger heterotrophic, than autotrophic, microbial response
in shallow euphotic sediments (Patching and Rose, 1970; Vázquez-Domínguez et al., 2012; Yang et al., 2016), and as such,
there would likely be more DOC remineralisation (Lønborg et al., 2018) than 'new' DOC production (Wohlers et al., 2009;
85 Engel et al., 2011; Novak et al., 2018)." The focus on euphotic sediment is also now made clear in the methods, LN 95:
"Sediment at the site was unvegetated and characterised as a euphotic cohesive sandy mud…" (with underlined sections
adjusted for clarity)

**(2)** As per the reviewer's suggestion, we will rearrange the last part of the introduction and include the recommended additions
in the structure, as follows:

90 **1 – dominating mechanisms acting on DOC:**
LN 64: "Primary producers fix DIC during photosynthesis and release DOC directly through exudation and/or indirectly when
they are grazed upon. Photosynthetically produced DOC is the main source of DOC in the ocean (Hansell et al., 2009). DOC

fuels local microbial mineralisation (Azam, 1998). Heterotrophic bacteria respire the carbon from DOC as $CO_2$, which can then be recaptured by photoautotrophs (Riekenberg et al., 2018), closing the microbial loop (Azam, 1998). DOC and DIC that is not captured is ultimately effluxed to the overlying water column and may be transported from estuaries to the coastal ocean." (with underlined sections added for clarity)

**2 – how warming and OA may affect these mechanisms:**

LN 69: "Individually, increased temperature and $CO_2$ can enhance primary productivity, and therefore DOC production, in arctic (Engel et al., 2013; Czerny et al., 2013) and temperate phytoplankton communities (Wohlers et al., 2009; Engel et al., 2011; Liu et al., 2017; Novak et al., 2018; Taucher et al., 2012), and temperate stream sediments (Duan and Kaushal, 2013). However, one study in a temperate fjord reported no enhancement of DOC production despite $CO_2$ enhanced phytoplankton productivity (Schulz et al., 2017). This uncertainty of response to individual climate stressors is exacerbated when considering how the combination of OA and warming will affect the production and degradation of DOC. To date, only one study has considered this combined stressor effect on DOC fluxes (Sett et al., 2018), observing no difference in DOC production by temperate phytoplankton relative to current conditions (Sett et al., 2018)."

**3 – Experimental design and hypotheses:**

LN 77: "To understand the potential effect of future climate on DOC fluxes, it is essential that both individual and combined effects of OA and warming are considered. Here we focus on changes in DOC fluxes in unvegetated estuarine sediments, as these systems have the potential for significant uptake of DOC that is currently exported to the coastal ocean. In this study, benthic DOC responses in unvegetated estuarine sediments were investigated over an 8 °C temperature range under both current and projected future high-$p$CO$_2$ conditions in an ex situ laboratory incubation." (with underlined sections adjusted for clarity)

LN 82: "We expected that warming would promote a stronger heterotrophic, than autotrophic, microbial response in shallow euphotic sediments (Patching and Rose, 1970; Vázquez-Domínguez et al., 2012; Yang et al., 2016), and as such, more DOC remineralisation (Lønborg et al., 2018) than 'new' DOC production (Wohlers et al., 2009; Engel et al., 2011; Novak et al., 2018)." (with underlined sections adjusted for clarity)

LN 85: "Moreover, despite the potential stimulation of primary productivity in unvegetated muddy sediments by OA (Vopel et al., 2018) or more likely high-$p$CO$_2$ availability, and potential enhancement of DOC production (Engel et al., 2013; Liu et al., 2017), this increase in labile DOC may promote bacterial productivity and DOC mineralisation (Hardison et al., 2013). In addition, increased DOC availability alone may increase heterotrophic bacterial biomass production and activity (Engel et al., 2013). We therefore predicted that increases in DOC production from OA alone or in combination with warming may be counteracted by increased consumer activity, potentially diminishing the available DOC pool under future climate conditions." (with underlined sections adjusted for clarity)

**Comment:** What influence would variable light conditions have on your findings? The cores are taken from a shallow estuarine site where one can expect considerable resuspension from tides, currents and winds. The light intensities used here are likely representative of best case. So, one can maybe amplify the dark scenario?

**Reply:** This is an interesting question that would be of interest to the general readership. We see value in addressing this question within the discussion and follow the same thought process as Reviewer 1, where the dark scenario responses would likely be amplified. The following sentence will be added, LN 339: "Under conditions of reduced light availability/intensity, sediments are expected to have an amplified heterotrophic response in addition to a reduction in microalgal production of DOC."

**Comment:** Line 7. "Estuaries make a disproportionately". What do you mean here? With respect to what?

**Reply:** This was unclear, the statement will be adjusted to read LN 7: "Relative to their surface area, estuaries make a disproportionately large contribution of dissolved organic carbon (DOC) to the global carbon cycle, but it is unknown how this will change under a future climate." (with underlined sections adjusted for clarity)

**Comment:** Line 19. DOC is smaller than that retained in soils and also in fossil fuels.

**Reply:** While this statement by reviewer 1 is valid, we do not believe what we said is untrue, LN 20: "The aquatic dissolved organic carbon (DOC) pool is one of the largest pools of organic carbon on earth (Hedges, 1987) and roughly equivalent in size to the atmospheric $CO_2$ reservoir (Siegenthaler and Sarmiento, 1993)." We do not say it is the largest, just one of the largest. For this reason, we intend to leave this sentence unchanged.

**Comment:** **(1)** Line 28. And **(2)** line 32-35. Here you state that 33% of the NPP in coastal waters is exported to the oceans and stored in the ocean interior. I question the validity of this statement/citation. **(3)** Is there evidence that the interior ocean is increasing in DOC? Why the large difference between mineralisation efficiency of DOC produced in surface water of the ocean to that produced in coastal waters?

**Reply:** **(1)** The line reads LN 29: "up to 33 % of the associated DOC is exported offshore and stored in the ocean interior". This upper value is based on Krause-Jensen and Duarte (2016) who found that substantial macroalgal DOC produced in the coastal zone and exported offshore was subducted below the mixed layer into the ocean interior (117 (36-194) Tg-C y$^{-1}$). The text can be adjusted for clarity, to avoid confusion that the 33% of NPP carbon reaches the ocean interior. The text will now read, LN 28: "The shallow coastal zone accounts for 1 to 10 % of global net primary production (NPP) (Duarte and Cebrián, 1996), with up to 33 % of the associated DOC exported offshore and reaching the ocean interior (Krause-Jensen and Duarte, 2016)." (with underlined sections adjusted for clarity)

**(2)** There was a lack of information in this paragraph regarding how the value of 3.5× was calculated. The paragraph now reads, LN 31: "Although shallow estuaries and fringing wetlands make up only ~22 % of the world's coastal area (Costanza et al., 1997) and 8.5 % of the total marine area (Costanza et al., 1997) they are quantitatively significant in terms of DOC processing and offshore transport (Smith and Hollibaugh, 1993). In 1998, Bauer and Druffel used radioisotopic carbon ($^{14}$C) to identify the source and age of DOC and POC inputs into the open ocean interior. They found that ocean margins accounted for greater organic carbon inputs into the ocean interior than the surface ocean by more than an order of magnitude. Assuming 1/3 of the DOC produced in the coastal zone (100-1900 Tg-C y$^{-1}$, Duarte, 2017) is subducted and reaches the ocean interior (Krause-Jensen and Duarte, 2016), 30 to 630 Tg-C y$^{-1}$, or up to 3.5× more DOC could reach the ocean interior from coastal areas than from the open ocean (180 Tg-C y$^{-1}$, Hansell et al., 2009). This is despite coastal areas having a DOC production rate

only 0.2 to 3.9 % that of the open ocean (Duarte, 2017). As such, small changes to the coastal production and export of DOC may have a disproportionate influence on the global DOC budget." (with underlined sections adjusted for clarity)

**(3)** We are not trying to suggest that the interior ocean DOC pool is increasing, but instead, that a disproportionally large amount of DOC in the interior ocean could be sourced from the coastal zone relative to the surface ocean. This is based on previous work looking into the transport of DOC from the coastal zone and surface ocean to the ocean interior, respectively (calculations detailed in (2)).

We have included in our introduction the following text to further support the importance and potential significance of changing the supply of coastal DOC to the ocean.

LN 33: "In 1998, Bauer and Druffel used radioisotopic carbon ($^{14}$C) to identify the source and age of DOC and POC inputs into the open ocean interior. They found that ocean margins accounted for greater organic carbon inputs into the ocean interior than the surface ocean by more than an order of magnitude."

**Comment:** Line 43. Delete extra "lability"

**Reply:** Thank you. This has been rewritten to avoid repeating "lability". LN 47: "These heterotrophic bacteria not only consume autochthonous DOC (Boto et al., 1989), but their biomass is influenced by the lability of sediment organic matter (OM) (Hardison et al., 2013), which can be directly linked to and stimulated by MPB (Hardison et al., 2013; Cook et al., 2007)." (with underlined sections adjusted for clarity)

**Comment:** First three paragraphs contradict. You start by arguing that coastal waters are an important source of DOC to the open ocean but then finish by stating that coastal sediments are an important sink for DOC.

**Reply:** This can be clarified by exaggerating the distinction between coastal zone as a whole and estuarine sediments as a part of that whole in the third paragraph. The intention is to highlight that the coastal zone is an important source of DOC for the global ocean, however in sediments heterotrophic bacteria can make unvegetated estuarine sediments a sink of DOC produced elsewhere. As such, it is important to assess the role of this potential sink under conditions of warming and OA. The third paragraph has therefore been adjusted below:

LN 41: "Euphotic estuarine sediments occupy the coastal boundary between terrestrial and marine ecosystems. Microalgal communities (microphytobenthos, or MPB) are ubiquitous in these sediments, occupying ~40 to 48 % of the coastal surface area (Gattuso et al., 2020), and generating up to 50 % of total estuarine primary productivity (Heip et al., 1995; MacIntyre et al., 1996; Underwood and Kromkamp, 1999). MPB exude some of the carbon they fix as extracellular substances, including carbohydrates (Oakes et al. 2010), and can therefore be a source of relatively labile DOC in net autotrophic sediments (Cook et al., 2004; Oakes and Eyre, 2014; Maher and Eyre, 2010). However, microbial mineralisation by heterotrophic bacteria (Azam, 1998) within the sediment communities are a dominant sink of DOC in coastal sediments (Boto et al., 1989). These heterotrophic bacteria not only consume autochthonous DOC from upstream (Boto et al., 1989), but their biomass is influenced by the lability of sediment organic matter (OM) (Hardison et al., 2013), which can be directly linked to and stimulated by MPB (Hardison et al., 2013; Cook et al., 2007). As such, estuarine sediments are a potentially important sink for DOC." (with underlined sections adjusted for clarity)

195    **Comment:** Line 48. Check referencing. Fischot and Benner paper does not address the processing of DOC by estuarine sediments.

   **Reply:** This is true. Fichot and Benner (2014) looks at shelf processes, not estuarine. However, it is likely that the euphotic unvegetated shelf sediments in Fichot and Benner (2014) would not be dissimilar to euphotic unvegetated estuarine sediments. A more nearshore reference would be by Sandberg et al. (2004), who found that tDOC was the dominant carbon source for

200    bacterial secondary production in the water column of Ore Estuary (Northern Baltic Sea).

   This has been reworded in the text as follows:

   LN 51: "Unvegetated estuarine sediments can affect the quantity and quality of DOC input to the ocean by 1) acting as a source of autochthonous DOC, through MPB production (Duarte, 2017; Krause-Jensen and Duarte, 2016; Maher and Eyre, 2010), or 2) modifying allochthonous and terrigenous DOC inputs (Fichot and Benner, 2014). Through efficient mineralisation of DOC

205    (Opsahl and Benner, 1997), estuaries can act as a sink for DOC and a source of $CO_2$ to the ocean (Frankignoulle et al., 1998; Fichot and Benner, 2014; Sandberg et al., 2004)." (with underlined sections adjusted for clarity)

   **Comment:** Line 55-60. The increased DOC production in the Engel et al 2013 study was due to nutrient limitation. When they added nutrients, it was rapidly removed again. So, no net accumulation of DOC.

   **Reply:** This reference will be removed from this section.

210    **Comment:** Line 287-289. This can be deleted.

   **Reply:** Agreed, it will be deleted.

   **Comment:** Line 340-343. Check phrasing and possible break into two sentences to make easier reading.

   **Reply:** The sentence has been adjusted for clarity. LN 350: "As well as differences in diffusive versus advective modes of solute transfer between the sediment types (Cook and Røy, 2006), differences may be partially due to sandier sediments being

215    limited by other factors such as nutrient and OM availability, given that coarser sediments are generally more oligotrophic (Admiraal, 1984; Heip et al., 1995)." (with underlined sections adjusted for clarity)

   **Comment:** Line 350-359. Here the authors begin to speculate about the lability of DOC without any measurements to support it. I am not sure it is necessary.

   **Reply:** We see what Reviewer 1 is saying. This paragraph functions without that sentence. As such, it will be deleted.

220    **Comment:** Line 395. DOC is also produced continually from the detrital sediment POC. This contributes to dark DOC production.

   **Reply:** We will add this source of dark DOC in the discussion.

   LN 408: "Although DOC is mainly produced by photoautotrophs, DOC can be produced in the dark through, for example, chemodegradation of detrital organic carbon and cell lysis by viruses and during grazing (Carlson, 2002). "

225    **Comment:** Line 398-399. Are you inferring nutrient limitation in your set up? For now, I have assumed you had adequate nutrients.

**Reply:** There were no apparent N limitations in the present study, however, we were opening up the discussion to gauge what could happen if there was a limitation in nutrients. The responses to Reviewer 1's comments, detailed above, add extra clarity to the nutrient availability for the sediments.

230 **Comment:** Line 401. A very bold statement and the reference (Costanza) does not seem to support it. Please check.

**Reply:** The reference was incorrect. Explanation for how this was calculated will be provided in the introduction LN 35: "Assuming 1/3 of the DOC produced in the coastal zone (100-1900 Tg-C $y^{-1}$, Duarte, 2017) reaches the ocean interior (Krause-Jensen and Duarte, 2016), 30 to 630 Tg-C $y^{-1}$, or up to 3.5× more DOC could reach the ocean interior from coastal areas than from the open ocean (180 Tg-C $y^{-1}$, Hansell et al., 2009)." and this reference will now read, LN 417: "Up to 3.5× more DOC

235 reaches the ocean interior from coastal areas than the open ocean (Duarte, 2017; Krause-Jensen and Duarte, 2016; Hansell et al., 2009).

**Comment: (1)** Figures Error bars in the figure should go both plus and minus. **(2)** Check text in figure 4. Do you not mean aerobic respiration (with arrow pointing upwards)?

**Reply: (1)** The figures will be changed into box and whisker plots to show the full range of data. This will satisfy Reviewer 1

240 and 2's concerns.

**(2)** We thank Reviewer 1 for catching this oversight. The arrows that are now on the figure are accurate.

**Anonymous Referee #2

This is a well described experimental case study that contributes to close an important knowledge gap concerning the modification of the carbon cycle under global environmental and climatic change. My biggest concern in the study is the

245 upscaling to the global dimension. The authors are aware of the associated risks and that such an upscaling may be (at least) quantitatively quite problematic. Overall, this is a thoroughly made study and a useful addition in the field.

Suggestions for a revised manuscript:

**Comment:** Section 4.3.3: The authors are correct in being very careful when they provide a daring global upscaling here. It would be good to add a paragraph on detailing why such an upscaling can be risky and possibly incorrect (different

250 hydrodynamic settings, different sediment composition, different delivery of dissolved and particulate matter from land and through aeolian deposition, etc.)

**Reply:** We agree. This section is highly speculative and is purely an exercise of interest, a likely exercise that readers will do on their own. We will follow Reviewer 2's suggestion and add further details regarding the limitations of the upscaling. Also, see our reply to Reviewer 1's comments.

255 **Comment:** Line 54: It is not only the climate project models but rather the scenarios used for the projections. The scenarios are usually produced through simplified climate models and integrated assessment models.

**Reply:** Yes, this is true. We had included the scenario reference at the end of the sentence (RCP8.5), however, it would be more forthcoming to include the "high-emission scenario climate projections" explicitly in the text. This adjustment will be added.

260     LN 59: "Climate projection models assuming a high-emission scenario suggest that atmospheric $CO_2$ concentrations could more than double by the end of the century, increasing the partial pressure of $CO_2$ ($pCO_2$) in surface waters to 1000 $\mu$atm and decreasing pH by 0.3 units, together termed ocean acidification (OA) (RCP8.5, IPCC, 2019)."

    **Comment:** Line 55: "increasing the partial pressure by 580 ppm" – relative to which reference year?

    **Reply:** This has been rewritten for clarity. LN 59: "$CO_2$ concentrations could more than double by the end of the century,
265     increasing the partial pressure of $CO_2$ ($pCO_2$) in surface waters to 1000 $\mu$atm …"

    **Comment:** Lines 55-60: Though regional primary production may be enhanced with temperature and $pCO_2$, climate change can lead to increased stratification and a decrease of mixing as well. It would be good to also discuss this aspect and cite a few relevant literature sources.

    **Reply:** This discussion of the possible effect of stratification will be added to the discussion section with the following text:
270     LN 428:" For example, the response to warming and $pCO_2$ may be different for pelagic communities and/or in deeper waters that are subject to stratification (Li et al., 2020), where access to nutrients and $CO_2$ may become limiting (Rost et al., 2008)."

    **Comment**: Line 140: "refit from Mehrbach et al. (1973)" – can you describe in more detail how and why you did this?

    **Reply:** We did not do the refit, Dickson and Millero (1987) did. The sentence reads, "Total borate concentrations (Uppström, 1974) and boric acid (Dickson, 1990) and stoichiometric equilibrium constants for carbonic acid (Dickson and Millero, 1987),
275     refit from Mehrbach et al. (1973), were used." We just wanted to include the original source of Dickson and Millero (1987). For clarity, this has been rewritten as LN 151: "…carbonic acid from Mehrbach et al. (1973) as refit by Dickson and Millero (1987), were used."

    **Comment:** Line 277: "OA alone (at ambient temperatures)" – what is meant with 'ambient temperatures' exactly?

    **Reply:** At ambient temperatures was meant to distinguish the OA scenario from the OA and temperature manipulation
280     scenarios. This would therefore be at 23 °C. This sentence would be improved with the addition of the temperature included. The text will now read, LN 220: "High-$pCO_2$ alone (at mean ambient temperatures, 23 °C)"

    **Comment:** Section headings "4.2 OA increases DOC uptake" and "4.3.2 Warming increases respiration and DOC uptake" are unclear. Which component takes up DOC? Maybe use a different word for 'uptake'?

    **Reply:** We can see the ambiguity in uptake. We believe assimilation would be a more accurate term as the heterotrophs in the
285     sediments actively assimilate DOC. The section headings will now read: LN 341: "4.2 OA increases DOC assimilation" and LN 367: "4.3 Warming drives increased heterotrophy and DOC assimilation" and LN 395: "4.3.2 Warming increases respiration and DOC assimilation"

    **Comment:** Figure 1: Some fonts are so tiny that they are not readable. Please, increase them if relevant or delete unnecessary information.

290     **Reply:** This will be adjusted as suggested.

    **Comment:** Figure 5: The 'bars' within the grey and dotted areas of the plot are barely visible. What do these 'bars' show? Please, provide information in the figure caption.

**Reply:** The figure has been redesigned. The figure caption will clearly indicate "
[revised manuscript text omitted]